Natural history and molecular evolution of demersal Mediterranean sharks and skates inferred by comparative phylogeographic and demographic analyses

Ferrari Alice 1
Tinti Fausto fausto.tinti@unibo.it 1
Bertucci Maresca Victoria 1 2
Velonà Alessandro 1
Cannas Rita 3
Thasitis Ioannis 4
Costa Filipe Oliveira 5
Follesa Maria Cristina 3
Golani Daniel 6
Hemida Farid 7
Helyar Sarah J. 8
Mancusi Cecilia 9
Mulas Antonello 3
Serena Fabrizio 10
Sion Letizia 11
Stagioni Marco 1
Cariani Alessia 1
1 Department of Biological, Geological & Environmental Sciences (BiGeA), University of Bologna , Bologna , Italy
2 Department of Life Sciences, University of Trieste , Trieste , Italy
3 Department of Life Sciences and Environment, University of Cagliari , Cagliari , Italy
4 Department of Fisheries and Marine Research, Ministry of Agriculture, Natural Resources and Environment , Nicosia , Cyprus
5 Centre of Molecular and Environmental Biology (CBMA), University of Minho , Braga , Portugal
6 Department of Evolution, Systematics and Ecology, Hebrew University of Jerusalem , Jerusalem , Israel
7 Ecole Nationale Supérieure des Sciences de la Mer et de Aménagement du Littoral (ENSSMAL) , Algiers , Algeria
8 School of Biological Sciences, Institute for Global Food Security, The Queen’s University Belfast , Belfast , United Kingdom
9 Regional Agency for Environmental Protection-Toscana (ARPAT) , Livorno , Italy
10 Institute Coastal Marine Environment, Italian National Research Council (CNR-IAMC) , Mazara del Vallo , Italy
11 Department of Biology, University of Bari , Bari , Italy
Amorim Antonio
Electronic publication date: 2018 Sep 18
Publication date: 2018
Volume: 6
Electronic Location ID: e5560
Received 2018 Mar 9; Accepted 2018 Aug 9
Copyright: ©2018 Ferrari et al.
Copyright year: 2018
Copyright holder: Ferrari et al.
License: This is an open access article distributed under the terms of the Creative Commons Attribution License, which permits unrestricted use, distribution, reproduction and adaptation in any medium and for any purpose provided that it is properly attributed. For attribution, the original author(s), title, publication source (PeerJ) and either DOI or URL of the article must be cited.
License URL: https://creativecommons.org/licenses/by/4.0/

Keywords: Chondrichthyans, Phylogeography, Demography, Natural history, Demersal elasmobranchs, Mediterranean sea, Sicilian channel, Catsharks, Skates, North-Eastern Atlantic Ocean

Funding: RFO Canziani FFO This research was funded by RFO and Canziani grants given to Fausto Tinti and Alessia Cariani, and by the FFO grant of the University of Bologna for the research fellowships of Alice Ferrari and Alessandro Velonà. The funders had no role in study design, data collection and analysis, decision to publish, or preparation of the manuscript.

==============================
Background

The unique and complex paleoclimatic and paleogeographic events which affected the Mediterranean Sea since late Miocene deeply influenced the distribution and evolution of marine organisms and shaped their genetic structure. Following the Messinian salinity crisis and the sea-level fluctuations during the Pleistocene, several Mediterranean marine species developed deep genetic differentiation, and some underwent rapid radiation. Here, we consider two of the most prioritized groups for conservation in the light of their evolutionary history: sharks and rays (elasmobranchs). This paper deals with a comparative multispecies analysis of phylogeographic structure and historical demography in two pairs of sympatric, phylogenetically- and ecologically-related elasmobranchs, two scyliorhinid catsharks (Galeus melastomus, Scyliorhinus canicula) and two rajid skates (Raja clavata, Raja miraletus). Sampling and experimental analyses were designed to primarily test if the Sicilian Channel can be considered as effective eco-physiological barrier for Mediterranean demersal sympatric elasmobranchs.

Methods

The phylogeography and the historical demography of target species were inferred by analysing the nucleotide variation of three mitochondrial DNA markers (i.e., partial sequence of COI, NADH2 and CR) obtained from a total of 248 individuals sampled in the Western and Eastern Mediterranean Sea as well as in the adjacent northeastern Atlantic Ocean. Phylogeographic analysis was performed by haplotype networking and testing spatial genetic differentiation of samples (i.e., analysis of molecular variance and of principal components). Demographic history of Mediterranean populations was reconstructed using mismatch distribution and Bayesian Skyline Plot analyses.

Results

No spatial genetic differentiation was identified in either catshark species, while phylogeographic structure of lineages was identified in both skates, with R. miraletus more structured than R. clavata. However, such structuring of skate lineages was not consistent with the separation between Western and Eastern Mediterranean. Sudden demographic expansions occurred synchronously during the upper Pleistocene (40,000–60,000 years ago) in both skates and G. melastomus, likely related to optimal environmental conditions. In contrast, S. canicula experienced a slow and constant increase in population size over the last 350,000 years.

Discussion

The comparative analysis of phylogeographic and historical demographic patterns for the Mediterranean populations of these elasmobranchs reveals that historical phylogeographic breaks have not had a large impact on their microevolution. We hypothesize that interactions between environmental and ecological/physiological traits may have been the driving force in the microevolution of these demersal elasmobranch species in the Mediterranean rather than oceanographic barriers.

Introduction

The Mediterranean Sea has been universally recognised as a cradle of biodiversity (Cuttelod et al., 2009; Coll et al., 2010; Lejeusne et al., 2010; Mouillot et al., 2011). This characteristic of the Mediterranean is due to its origin in the unique and complex paleoclimatic and paleogeographic histories of the ancient Paratethys (Rögl, 1999) and still relies on the unusual salinity and water circulation conditions, driven by topography and local climatic regimes (Robinson et al., 2001). After the almost total closure of the Atlantic seaway, the basin experienced a nearly complete desiccation (the Messinian Salinity Crisis, ∼5.33 MYA) and about 40 warm interglacial-cold glacial cycles during the Pleistocene (from 2.5 to 0.01 MYA), which caused sea-level oscillations and sea-water temperature changes (Waelbroeck et al., 2002). These events have deeply influenced the distribution and evolution of marine organisms and shaped their genetic structure (Nikula & Väinölä, 2003; Boudouresque, 2004; Duran, Pascual & Turon, 2004; Wörheide, Solé-Cava & Hooper, 2005; Pérez-Losada et al., 2007).

The majority of genetic studies on Mediterranean fishes have focussed on teleosts (Magoulas et al., 2006; Rolland et al., 2007), whilst there have been far fewer studies conducted on elasmobranchs (Chevolot et al., 2006; Griffiths et al., 2010; Iglésias, Toulhoat & Sellos, 2010; Pasolini et al., 2011). Considering their uniqueness in terms of evolutionary history, Mediterranean elasmobranchs (89 species; FAO, 2018a; FAO, 2018b) should be high priority for conservation plans to protect against overfishing, by-catch, habitat loss and fragmentation (Serena, 2005; Cavanagh & Gibson, 2007; Serena, Mancusi & Barone, 2010; Froese & Pauly, 2017; Stein et al., 2018). Hence, for conservation and management purposes, genetic surveys have recently been carried out to disentangle the genetic structure, phylogeography and gene flow among several Mediterranean populations in an important fishery resources, including Scyliorhinus canicula (Barbieri et al., 2014; Gubili et al., 2014; Kousteni et al., 2015) and the endemic skates Raja polystigma (Frodella et al., 2016) and R. asterias (Cariani et al., 2017).

By analysing mtDNA variation in sympatric species and reconstructing the impact of the past events and processes leading to contemporary biota, comparative phylogeography can contribute to inferences of common evolutionary and demographic processes (Avise et al., 1987; Avise, 2000; Arbogast & Kenagy, 2001). In particular, phylogeography helps to unravel the distribution of ancestral lineages based on haplotypes shared by contemporary individuals under a coalescence process. Moreover, targeting phylogenetically and ecologically closely-related species may help to identify unifying/similar mechanisms triggering the evolution of particular marine species (Arbogast & Kenagy, 2001). Hence, the coupling of a phylogeographic approach with historical demography may empower the testing of micro-evolutionary hypotheses (Drummond et al., 2005; Campos et al., 2010; Ho & Shapiro, 2011), the identification of factors driving past population dynamics (Finlay et al., 2007; Patarnello, Volckaert & Castilho, 2007; Atkinson, Gray & Drummond, 2008; Stiller et al., 2010) and range expansions (Fahey et al., 2012).

Within the Mediterranean Sea, the transition area of the Sicilian Channel has been considered a major barrier to the dispersal of marine species between the Western and Eastern sub-basins, even if it is not the unique barrier assessed therein (Bianchi & Morri, 2000; Bianchi, 2007; Patarnello, Volckaert & Castilho, 2007; Coll et al., 2010). However, the role of the Sicilian Channel as a partial barrier to marine species dispersal rather than a transition/genetic admixture area is still under scrutiny (Pascual et al., 2017). This transition area affects the species richness of elasmobranchs, which are higher in the western part of the Mediterranean than in the eastern part (Coll et al., 2010). However, even if tested with different experimental designs and molecular markers, the barrier role of the Sicilian Channel transition area in the geographical structuring of mtDNA variation seems to be comparatively low. The small-spotted catshark S. canicula exhibited strong genetic structure as revealed by significant mtDNA and microsatellite-based fixation indexes, regardless a consistent strong phylogeographic break between western and eastern populations. Indeed, haplotypes from the two sub-basins were phylogenetically intermingled and weakly divergent in the haplotype median-joining networks (Barbieri et al., 2014; Gubili et al., 2014; Kousteni et al., 2015). In R. polystigma, Frodella et al. (2016) found a lack of genetic structure and a very weak phylogeographic break between the western Mediterranean and the Adriatic population, representing the only sample from the eastern sub-basin.

The processes shaping the genetic architecture in marine species are affected by historical abundance and dispersal. Changes in population size and geographical distribution can be reflected by changes in the genetic diversity and differentiation (Grant & Waples, 2000; Patarnello, Volckaert & Castilho, 2007). Although temporal estimates based on molecular-clock calibration are constrained by molecular and physiological parameters (due to differences in evolutionary rates across taxa, and the effects of differences in generation time, metabolic rate and body size; Martin & Palumbi, 1993; Gillooly et al., 2005; Grant et al., 2012), haplotype diversity and divergence still demonstrate different signatures in bottlenecked, expanding and constant-size populations (Grant & Waples, 2000; Patarnello, Volckaert & Castilho, 2007). Improved models and analytical tools for the inference of demographic changes over time based on genetic data have been shown to be highly informative for elucidating past population dynamics (Kuhner, 2009). It has been demonstrated that the elasmobranch mtDNA substitution rate is approximately 10% slower than that of teleosts, and that the combined use of multiple sequence markers can improve the resolution of phylogeographic and demographic analyses (Frodella et al., 2016).

This paper deals with a comparative multispecies analysis of phylogeographic structure and historical demography in two pairs of sympatric, phylogenetically- and ecologically- related elasmobranchs to test for common natural histories and environmental/climatic factors (i.e., phylogeographic breaks), which may potentially have driven their microevolution. Nucleotide variation in the target species, namely the rajid skates Raja clavata L. (thornback ray) and R. miraletus L. (brown skate), and the scyliorhinid catsharks Galeus melastomus (Rafinesque, 1810; blackmouth catshark) and Scyliorhinus canicula L. (small-spotted catshark), was analysed at three mitochondrial gene fragments which have been proven polymorphic at the population level (Barbieri et al., 2014; Gubili et al., 2014; Kousteni et al., 2015; Frodella et al., 2016; Cariani et al., 2017; Ramírez-Amaro et al., 2018). Sampling and experimental analyses were designed to primarily test if the Sicilian Channel has been acted as an eco-physiological barrier for Mediterranean demersal sympatric elasmobranchs. Moreover, the inclusion of several population samples from two geographical areas within each sub-basin will allow the opportunity to detect additional phylogeographic breaks in the region.

Materials and Methods

Sampling

A total of 248 tissue specimens (fin clip or skeletal muscle) and associated biological data were collected during international research cruises (e.g., MEDITS Bertrand et al., 2002) or by contracted commercial fishermen between 2001 and 2010 from 17 sites, located in the northeastern Atlantic and throughout the Mediterranean Sea (Fig. 1, Table 1, Table S1). The sampling design was established according to the main zoogeographical boundary dividing Western (WMED) and Eastern (EMED) Mediterranean sub-basins as proposed by Pérès & Picard (1964), Giaccone & Sortino (1974) and Bianchi & Morri (2000), Bianchi (2007). Within each sub-basin, sampling locations were grouped into two geographical areas (Table 1, Table S1). In this study, satisfactory sample sizes remained a major challenge, especially for the species that are inadequately represented in commercial catches (i.e., R. miraletus) due to catchability and/or selectivity characteristics of associated fishing methods (i.e., trawl system vs. “rapido” trawl). Individuals were specifically assigned using the available identification guidelines and keys (Serena, 2005; Serena, Mancusi & Barone, 2010).

Figure 1 Sampling locations of the four demersal elasmobranchs in the study.

Numbers of sampling locations refer to Table 1. Sampling locations for each species are colour coded according to four geographical areas as reported in Table 1. North-Eastern Atlantic sampling locations are not represented. The map was created using R v.3.4.1 (R Core Team, 2016; Wickham, 2010; Becker et al., 2016).

Table 1 Sampling data.

Sampling information, including species, sub-basin and geographical area, sampling location and country, sampling code (see Fig. 1 for location), sample size (N), sampling year and geographical coordinates of the Mediterranean and North-eastern Atlantic samples of the four target elasmobranch species.

Species	Sub-basin/ Geographical area	Sampling location, Country	Sample code	N	Sampling year	Latitude (N)	Longitude (E)	
Raja clavata	Western Mediterranean							
	Algerian coasts	Annaba, Algeria	1	5	2001	36°54′166	07°47′152	
		Bouharoun, Algeria	3	6	2003	36°37′472	02°38′124	
	Ligurian-Tyrrhenian Sea	Genova, Italy	5	3	2001	44°22′171	08°50′152	
		Fiumicino, Italy	6	4	2001	41°45′273	12°07′324	
		Tuscany, Italy	7	7	2002	43°22′022	09°55′322	
		Cagliari, Italy	8	3	2009	38°52′011	09°20′353	
	Eastern Mediterranean							
	Adriatic Sea	Marche, Italy	10	5	2002	43°21′330	14°17′510	
		Croatian coasts, Croatia	11	3	2002	45°07′845	14°25′664	
		Fano, Italy	12	9	2006	44°31′120	13°00′250	
	Levantine Sea	Antakya, Turkey	14	6	2004	36°15′203	35°19′114	
		Iskenderun, Turkey	15	5	2004	35°52′352	33°33′532	
		Cyprus coasts, Cyprus	16	4	2009	34°21′396	33°08′564	
	North-Eastern Atlantic							
		Portuguese coasts, Portugal		8	2005	40°85′00	9°20′00	
		North Cardigan, UK		1	2006	52°42′00	4°32′00	
		The Wash, UK		1	2006	53°08′00	1°33′00	
		Total		70				
Raja miraletus	Western Mediterranean							
	Algerian coasts	Cap Djinet, Algeria	2	7	2009	36°53′012	03°40′201	
		Annaba, Algeria	1	3	2010	36°54′166	07°47′152	
		Tipaza, Algeria	4	2	2010	36°50′176	03°22′063	
	Balearic-Tyrrhenian Sea	Tuscany, Italy	7	8	2010	43°22′022	09°55′322	
		Southern Minorca, Spain	9	7	2008	39°35′540	04°33′306	
	Eastern Mediterranean							
	Adriatic Sea	Fano, Italy	12	11	2006	44°31′120	13°00′250	
		Croatian coasts, Croatia	11	9	2002	45°07′845	14°25′664	
	Levantine Sea	Haifa, Israel	17	4	2010	32°53′567	34°17′550	
		Iskenderun, Turkey	15	3	2004	35°52′352	33°33′532	
		Cyprus coasts, Cyprus	16	3	2009	34°21′396	33°08′564	
	North-Eastern Atlantic							
		Portuguese coasts, Portugal		3	2005	38°08′456	08°49′579	
		Total		60				
Galeus melastomus	Western Mediterranean							
	Algerian coasts	Bouharoun, Algeria	3	19	2010	36°37′472	02°38′124	
	Tyrrhenian Sea	Cagliari, Italy	8	17	2009	38°52′011	09°20′353	
	Eastern Mediterranean							
	Adriatic-Ionian Sea	Fano, Italy	12	8	2007	44°31′120	13°00′250	
		Ionian Sea, Italy	13	4	2008	39°39′065	17°37′390	
	Levantine Sea	Cyprus coasts, Cyprus	16	5	2009	34°21′396	33°08′564	
	North-Eastern Atlantic							
		Rockhall Plateau, UK		5	2010	–	–	
		Total		58				
Scyliorhinus canicula	Western Mediterranean							
	Algerian coasts	Bouharoun, Algeria	3	10	2010	36°37′472	02°38′124	
		Annaba, Algeria	1	3	2010	36°54′166	07°47′152	
		Tipaza, Algeria	4	12	2010	36°50′176	03°22′063	
	Tyrrhenian Sea	Cagliari, Italy	8	14	2009	38°52′011	09°20′353	
	Eastern Mediterranean							
	Adriatic Sea	Fano, Italy	12	10	2006	44°31′120	13°00′250	
	Levantine Sea	Cyprus coasts, Cyprus	16	5	2009	34°21′396	33°08′564	
	North-Eastern Atlantic							
		Rockhall Plateau, UK		6	2010	–	–	
		Total		60				

Sampling of sharks and skates was carried out fully respecting the fishing dispositions of the Regulation of the European Parliament and of the Council on certain provisions for fishing in the GFCM (General Fisheries Commission for the Mediterranean) Agreement area and amending Council Regulation (EC) No 1967/2006 concerning management measures for the sustainable exploitation of fishery resources in the Mediterranean Sea, adopted by the Council on 20th October 2011 (2011/C 345 E/01).

Molecular methods

After on-board collection, individual tissues were preserved in 96% ethanol at −20 °C until laboratory analyses. Total genomic DNA (gDNA) was extracted using the CTAB protocol (Doyle & Doyle, 1987). Three mitochondrial gene fragments were amplified and sequenced: the cytochrome oxidase c subunit I (COI), the nicotinamide dehydrogenase subunit 2 (NADH2) and the control region (CR). Skate-specific primer pairs for the amplification of the mitochondrial NADH2 gene were designed. This was carried out using homologous complete mitochondrial DNA (mtDNA) sequences of Okamejei kenojei (AY525783.1; NC_007173.1; Kim et al., 2005) and Amblyraja radiata (NC_000893.1; Rasmussen & Arnason, 1999). Available sequences were retrieved from GenBank and aligned using ClustalW algorithm implemented in MEGA 7 (Kumar, Stecher & Tamura, 2016). A consensus sequence was then used as input for primer design. Primer pairs were chosen using the online tool PRIMER3 v.4 (Untergasser et al., 2012) according to the minimum probability of primers to produce dimers or hairpins. Primers were tested on a Biometra Gradient Thermocycler to define the most suitable melting temperatures (Tm ranging from 50 °C to 60 °C). The complete list of primers used to amplify mtDNA gene fragments in each species is reported in Table S2. PCR reactions were performed in 50 µL reactions using the Taq DNA polymerase PCR kit (Invitrogen). The thermal profile consisted of an initial denaturation step at 94 °C for 5 min, 35 cycles of denaturation at 94 °C for 30 s, annealing at Ta (as detailed in Table S2) for 30 s, extension at 72 °C for 30 s, and a final elongation step at 72 °C for 10 min. Total PCR products were purified and sequenced at Macrogen Europe.

Data analyses

For each species, COI, NADH2 and CR partial sequences were aligned with ClustalW algorithm implemented in MEGA and concatenated to generate a combined dataset for subsequent analyses.

The software DnaSP v.6 (Rozas et al., 2017) was used to identify the number of haplotypes, the number of polymorphic and parsimony informative sites for each mitochondrial marker and for the combined dataset. Haplotype and nucleotide diversity (±standard deviations) were computed as measures of genetic diversity for each geographical area and sub-basin, and significant differences between the two sub-basins were tested for with Ruxton (2006) test.

Species-specific haplotype networks were created using the TCS method as implemented in PopART v.1.7 (Leigh & Bryant, 2015).

The Analysis of Molecular Variance (AMOVA) was performed with Arlequin v.3.5 (Excoffier & Lischer, 2010) grouping the Mediterranean samples on the basis of a priori hierarchical geographical structure on three levels: between sub-basins; between geographical areas, within sub-basin; within geographical areas. The statistical significance of the resulting Φ values was estimated by comparing the observed distribution with a null distribution generated by 10,000 permutations, in which individuals were redistributed randomly into samples.

The population structure was also assessed by a Principal Component Analysis (PCA) performed with the software Past v.2.03 (Hammer, Harper & Ryan, 2001) on a pairwise genetic distance matrix calculated between individuals using the best-fitting models selected with MEGA. The 95% ellipses were plotted to obtain the probabilistic distribution space of each geographical population sample.

The demographic histories of the Mediterranean shark and skate populations were reconstructed using three different approaches. We firstly performed the Tajima’s D, Fu’s FS and Ramos-Onsis & Rozas’s R2 neutrality tests (Tajima, 1989; Fu, 1997; Ramos-Onsins & Rozas, 2002) as implemented in DnaSP. Under a population expansion model, significant negative values of D and FS and significant positive values of R2were expected; the statistical significance was tested using 10,000 permutations. In the second approach, we estimated the “mismatch distribution” (Rogers & Harpending, 1992), i.e., the frequency distribution of the pairwise differences among sequences. The mismatch distribution was estimated under the assumption of a sudden expansion model as implemented in Arlequin. To determine the fit of our experimental data to the model distribution, the sum of squared deviations (SSD) between observed and expected mismatch distributions and the raggedness index (rg) were used as test statistics with 1,000 bootstrap replicates. Lastly, we reconstructed the historical demography using the coalescent-based Bayesian Skyline Plot approach (BSP; Kingman, 1982a; Kingman, 1982b; Drummond et al., 2005; Ho & Shapiro, 2011) implemented in the software package BEAST v.1.8.4 (Drummond et al., 2012), under the best-fit models previously selected, a strict molecular clock and a mutation rate of 0.005/million years (Chevolot et al., 2006). This mutation rate was obtained using a substitution rate estimated in a wider taxonomic framework using the times of divergence between Rajinae and Amblyrajinae (at 31 Myr) and within the main groups within Rajinae determined by Valsecchi et al. (2005). However, because mutation rates may vary across elasmobranch lineages and across genes within the mtDNA genome, applying this mutation rate to catsharks and to other mtDNA gene fragments should be considered with caution. To ensure convergence of the posterior distributions, we performed two independent Markov Chain Monte Carlo runs of 50,000,000 generations sampled every 5,000 generations, with the first 10% of the sampled points discarded as burn-in. Runs were subsequently combined in LogCombiner (included in the BEAST package). The quality of the run was assessed by effective sample size (ESS) >200 for each parameter using Tracer v.1.5 (Rambaut & Drummond, 2007). This software was also used to generate the skyline plots.

Results

The sequence variation of the separate and combined datasets in each species is reported in Table 2. As expected, the non-coding CR haplotypes (81) outnumbered those of the coding genes (COI: 38; NADH2: 43). The percentage of variable sites and of parsimony informative sites for the combined dataset ranged from 1.5% to 2.5% and from 0.7% to 1.8%, respectively.

Table 2 Sequence information of the separate and combined mtDNA datasets in the four target species.

Species	COI	NADH2	CR	Combined	
	SLH	NH	VS	Pi	SLH	NH	VS	Pi	SLH	NH	VS	Pi	SLH	NH	VS	Pi	
Raja clavata	579	9	16	7	748	8	8	4	354	9	7	2	1,681	26	31	13	
Raja miraletus	601	7	7	6	698	14	15	7	732	12	13	12	2,031	21	35	25	
Galeus melastomus	593	10	9	0	673	9	9	7	667	16	11	11	1,933	30	29	18	
Scyliorhinus canicula	596	12	13	6	749	12	11	10	624	44	25	21	1,969	54	49	37	
Total		38								81				131			
Notes.

SLH haplotype sequence length

NH number of haplotypes

VS number of variable sites

Pi number of parsimony informative sites

Estimates of haplotype diversity (h) were high in all samples, with the exception of the skate samples from Balearic, Ligurian and Tyrrhenian Seas (Table 3). Mean values of h and nucleotide diversity (π) were significantly higher in the EMED than in the WMED samples in all species, except for S. canicula (Ruxton test, P < 0.001; Table 3).

Table 3 Mitochondrial gene polymorphism of the species considered across the sampling areas.

Species	Geographical area/ Sub-basin	N	NH	h± S.D.	P	π± S.D.	P	
Raja clavata	North-Eastern Atlantic	10	5	0.756 ± 0.130		0.00241 ± 0.00066		
	Algerian coasts	11	6	0.727 ± 0.144		0.00054 ± 0.00015		
	Ligurian-Tyrrhenian Sea	17	4	0.493 ± 0.131		0.00032 ± 0.00010		
	Adriatic Sea	17	8	0.779 ± 0.099		0.00114 ± 0.00018		
	Levantine Sea	15	8	0.829 ± 0.085		0.00078 ± 0.00014		
	WMED	28	8	0.579 ± 0.104	<0.001	0.00041 ± 0.00009	<0.001	
	EMED	32	15	0.855 ± 0.050		0.00117 ± 0.00016		
Raja miraletus	North-Eastern Atlantic	3	3	1.000 ± 0.272		0.00098 ± 0.00035		
	Algerian coasts	12	7	0.773 ± 0.128		0.00081 ± 0.00023		
	Balearic- Tyrrhenian Sea	15	2	0.133 ± 0.112		0.00007 ± 0.00006		
	Adriatic Sea	20	4	0.574 ± 0.090		0.00209 ± 0.00025		
	Levantine Sea	10	6	0.844 ± 0.103		0.00373 ± 0.00071		
	WMED	27	9	0.698 ± 0.082	<0.001	0.00109 ± 0.00017	<0.001	
	EMED	30	10	0.798 ± 0.057		0.00323 ± 0.00034		
Galeus melastomus	North-Eastern Atlantic	5	4	0.900 ± 0.161		0.00083 ± 0.00025		
	Algerian coasts	19	13	0.942 ± 0.037		0.00231 ± 0.00022		
	Tyrrhenian Sea	17	10	0.882 ± 0.059		0.00196 ± 0.00033		
	Adriatic-Ionian Sea	12	12	1.000 ± 0.034		0.00247 ± 0.00037		
	Levantine Sea	5	5	1.000 ± 0.126		0.00321 ± 0.00077		
	WMED	36	17	0.916 ± 0.028	<0.001	0.00213 ± 0.00018	<0.001	
	EMED	16	16	0.993 ± 0.023		0.00269 ± 0.00034		
Scyliorhinus canicula	North-Eastern Atlantic	6	6	1.000 ± 0.096		0.00400 ± 0.00074		
	Algerian coasts	25	24	0.997 ± 0.012		0.00336 ± 0.00029		
	Tyrrhenian Sea	14	10	0.923 ± 0.060		0.00254 ± 0.00039		
	Adriatic Sea	10	10	1.000 ± 0.045		0.00326 ± 0.00046		
	Levantine Sea	5	5	1.000 ± 0.126		0.00284 ± 0.00055		
	WMED	39	34	0.989 ± 0.010	NS	0.00355 ± 0.00021	NS	
	EMED	15	15	1.000 ± 0.024		0.00340 ± 0.00033		
Notes.

N Sample size

NH number of haplotypes

h haplotype

p nucleotide diversity (±standard deviations, S.D.) for each species in each geographical area and sub-basin

P probability values of the Ruxton test between WMED vs. EMED hand p values

NS not significant

The thornback ray R. clavata showed a star-like network with the most common haplotype shared by all Mediterranean samples as well as by the northeastern Atlantic (NEATL) (Fig. 2). Slightly differentiated geographical haplotype lineages were detected in EMED (i.e., one formed by Adriatic individuals and one predominated by Levantine individuals) and in WMED. In addition to the most common haplotype, two low-frequency haplotypes were shared between Algerian coasts and the Adriatic and Levantine Sea. A divergent NEATL lineage formed by two unique haplotypes was also detected.

Figure 2 TCS network of combined haplotype of Raja clavata.

Circles are proportional to haplotype frequencies. Colours are consistent with Fig. 1; with the North-eastern Atlantic haplotypes indicated in green. Orthogonal bars between branch nodes indicate substitutions.Black nodes represent unsampled sequences.

In contrast, the brown skate R. miraletus, exhibited a stronger phylogeographic structure across the Mediterranean samples than R. clavata, indicated by the presence of three Mediterranean and one northeastern Atlantic/Western Mediterranean haplotype clusters (Fig. 3). Another lineage included haplotypes shared by Portuguese (NEATL) and Algerian individuals. This haplogroup slightly differed from a second haplogroup formed by the most frequent Mediterranean haplotype shared by Balearic-Tyrrhenian Sea and Adriatic Sea individuals and three single-individual haplotypes. Additionally, two more divergent lineages were constituted by Adriatic-Levantine and Levantine brown skates.

Figure 3 TCS network of combined haplotype of Raja miraletus.

Circles are proportional to haplotype frequencies. Colours are consistent with Fig. 1; with the North-eastern Atlantic haplotypes indicated in green. Orthogonal bars between branch nodes indicate substitutions. Black nodes represent unsampled sequences.

The haplotype networks of both catsharks showed a complete lack of phylogeographic signal, without any geographical distinction between Mediterranean and NEATL samples, or between WMED and EMED samples (Figs. 4 and 5).

Figure 4 TCS network of combined haplotype of Galeus melastomus.

Circles are proportional to haplotype frequencies. Colours are consistent with Fig. 1; with the North-eastern Atlantic haplotypes indicated in green. Orthogonal bars between branch nodes indicate substitutions. Black nodes represent unsampled sequences.

Figure 5 TCS network of combined haplotype of Scyliorhinus canicula.

Circles are proportional to haplotype frequencies. Coloursare consistent with Fig. 1; with the North-eastern Atlantic haplotypes indicated in green. Orthogonal bars between branch nodes indicate substitutions. Black nodes represent unsampled sequences.

The PCA results were consistent with those of the haplotype network analysis (Fig. 6), revealing a lack of spatial structure in G. melastomus and S. canicula, the partial genetic differentiation of NEATL and Adriatic samples of R. clavata and the separation of R. miraletus samples into three genetic groups.

Figure 6 Principal Components Analysis plots.

Plots illustrating the spatial structure of the elasmobranch species considered: (A) Raja clavata, (B) R. miraletus, (C) Galeus melastomus and (D) Scyliorhinus canicula. Colours are consistent with Fig. 1; the North-eastern Atlantic haplotypes are indicated in green. The PCA were carried out on individual sequences, however, only the haplotypes are reported here (dots). For each sample, the 95% ellipse illustrating the probabilistic distribution space of each geographical sample are shown.

The AMOVA (Table 4) did not reveal significant divergence between WMED and EMED in any species, even though R. miraletus and G. melastomus showed remarkable percentages of molecular variation between sub-basins (28.70% and 7.01%, respectively). Conversely, high molecular variation was detected between geographical areas within sub-basins in R. clavata, R. miraletus and S. canicula, revealing a genetic structure independent from their grouping in Western and Eastern Mediterranean sub-basins.

The estimated Tajima’s D, Fu’s FS and Ramos-Onsins & Rozas’s R2 indexes were largely consistent within each species (Table 5). Significant indexes strongly suggested a sudden demographic expansion of the Mediterranean R. clavata, while no significant values were obtained for the populations of other species with the exception of Fu’s FS index in S. canicula.

Both catsharks showed a unimodal mismatch distribution, R. clavata showed a skewed unimodal mismatch distribution, and the mismatch distribution of R. miraletus was bimodal (Fig. 7). The statistical analysis of the species-specific mismatch distributions (Table 5, SSD and rg indexes) revealed that the observed distributions were not statistically different from those expected under a sudden expansion model in all species, even if the SSD and rg values of R. miraletus were higher than those estimated for the other species (Table 5). In R. clavata, R. miraletus and G. melastomus the BSP analysis consistently indicated that sudden demographic expansions occurred approximately 40,000–60,000 years ago, while S. canicula exhibited a constant, slow demographic expansion over the last 350,000 years (Fig. 8).

Discussion

This study was characterized by a high sampling effort in terms of geographical coverage. Despite the variability and some restrictions of the sample size, we were able to sample 17 locations across the Mediterranean Sea as well as including samples from the northeastern Atlantic. One limitation of this work is the lack of samples from the Alboran and Aegean Seas, two areas that are known to be conservation and biodiversity hot spots for elasmobranchs (Coll et al., 2010) and in which some of the target species have shown independent historical population dynamics (e.g., the demographic decline shown by S. canicula in the Aegean Sea; Kousteni et al., 2015).

The multispecies comparative analysis did not reveal phylogeographical structuring on a latitudinal scale. However, distinct groups were found within sub-basins in both the brown skate and the small-spotted catshark, although not for the tested hypothesis of western-eastern historical separation. These results refute the hypothesis of an effective role of the Siculo-Tunisian transition area as a barrier in shaping natural history and microevolution of these demersal sharks and skates. However, it should be noted that physical and/or physiological barriers (e.g., the Strait of Gibraltar, the Almeria-Oran oceanographic front) did not always differentiate populations of marine phylogenetically- and ecologically-related taxa with similar differentiation patterns for instance, the discordant differentiation genetic patterns of the Atlantic-Mediterranean seabream species Diplodus puntazzo and D. sargus described in Bargelloni et al. (2005). In addition, the Siculo-Tunisian transition area can also affect the contemporary population connectivity and gene flow, leading to strong genetic divergence among western and eastern populations as shown for the small-spotted catshark S. canicula (Gubili et al., 2014; Kousteni et al., 2015).

Our results indicate a lack of deep phylogeographic structure in both scyliorhinid sharks: the blackmouth catshark G. melastomus, a species widely distributed in the outer continental shelves and upper slopes (55–1,200 m depth), and the small-spotted catshark S. canicula, which inhabits the shallow waters of continental shelves (prevalent in 10–100 m depth) and the uppermost slopes (200–400 m depth). For the blackmouth catshark, our findings are consistent with the results of a recent small-scale study, which showed an absence of population structure and high connectivity in the Western Mediterranean Sea (Ramírez-Amaro et al., 2018). In contrast, the lack of phylogeographic structure we observed in the Mediterranean for the small-spotted catshark could not be considered to be entirely consistent with the results of previous studies. Gubili et al. (2014) combined mitochondrial (CR) and nuclear (SSRs) markers and detected a strong differentiation within the Mediterranean Sea, where regardless of which molecular marker was used. Accordingly, the population from Eastern Mediterranean was significantly divergent from those of the other geographical areas (Gubili et al., 2014). Similarly, Kousteni et al. (2015) identified a deep genetic structure between the Western and the Eastern sub-basins and a weak differentiation of the Aegean population. In contrast, samples from the Levantine Sea shared haplotypes with both Western and Eastern sub-basins. Barbieri et al. (2014) analysed the geographical variation of S. canicula and observed several region-unique COI haplotypes corresponding to the Western-Eastern Mediterranean Sea and the Adriatic Sea, suggesting that the Sicilian Channel could not be considered a barrier to gene flow for this species. However, the haplotype networks built in all these studies did not detect western and eastern Mediterranean mtDNA clades, though all haplotype-frequency-based estimates indicated strong and significant contemporary differentiation between the sub-basins and, within each of them, among populations from different seas (Barbieri et al., 2014; Gubili et al., 2014; Kousteni et al., 2015). Such discrepancies could be due to incomplete lineage sorting likely related to the recent origin of the Mediterranean populations. In order to infer a reasonable phylogeographic scenario, even more Mediterranean areas need to be included, such as the Alboran and Aegean Seas, as well as samples from the southernmost coasts (e.g., Gulf of Sirte). Nevertheless, the phylogeographic pattern we detected in S. canicula widely overlaps with those identified in previous works.

Table 4 Analysis of Molecular Variance (AMOVA) of mtDNA of the four elasmobranch species.

Source of variation	d.f.	Sum of squares	% of variation	P	
Raja clavata					
Between sub-basins	1.00	3.05	−1.51	1.00	
Between areas, within sub-basin	2.00	6.57	22.85	0.00	
Within geographical areas	56.00	35.03	78.66	0.00	
Raja miraletus					
Between sub-basins	1.00	55.94	28.70	0.35	
Between areas, within sub-basin	2.00	39.51	32.87	0.00	
Within geographical areas	53.00	84.42	38.44	0.00	
Galeus melastomus					
Between sub-basins	1.00	6.23	7.01	0.32	
Between areas, within sub-basin	2.00	4.68	0.37	0.39	
Within geographical areas	48.00	109.11	92.61	0.08	
Scyliorhinus canicula					
Between sub-basins	1.00	9.99	−4.15	1.00	
Between areas, within sub-basin	2.00	27.49	23.18	0.00	
Within geographical areas	50.00	151.95	80.97	0.00	
Notes.

d.f. degrees of freedom

P probability values

Table 5 Historical demography of the target elasmobranch species.

The Tajima’s D, Fu’s FS, Ramos-Onsins & Rozas R2 and mismatch distributions indices (i.e., sum of squared deviations from the sudden expansion model, SSD, and raggedness index, rg) are reported for each species. The corresponding P-values are given in brackets.

Species	D	FS	R2	Mismatch distribution	
				SSD	rg	
Raja clavata	−2.260 (0.000)	−20.287 (0.000)	0.032 (0.002)	0.002 (0.320)	0.050 (0.540)	
Raja miraletus	−0.393 (0.376)	−2.724 (0.210)	0.090 (0.373)	0.026 (0.190)	0.054 (0.090)	
Galeus melastomus	−0.998 (0.179)	−18.111 (0.999)	0.072 (0.165)	0.004 (0.680)	0.016 (0.660)	
Scyliorhinus canicula	−1.110 (0.117)	−61.416 (0.000)	0.073 (0.166)	0.001 (0.540)	0.007 (0.710)	

Figure 7 mtDNA mismatch distribution of the four demersal elasmobranchs in the study.

Observed (bars) and expected (line) mismatch distributions under the sudden expansion model for the Mediterranean populations of the four target species. (A) Raja clavata. (B) Raja miraletus. (C) Galeus melastomus. (D) Scyliorhinus canicula. Results of the statistical analysis of the mismatch distributions are reported in Table 5.

Figure 8 Bayesian Skyline Plot of the four demersal elasmobranchs in the study.

Plot showing changes in the female effective population sizes (Neft) during time (MYA, million years ago) in the Mediterranean populations of the four target species. (A) Raja clavata. (B) Raja miraletus. (C) Galeus melastomus. (D) Scyliorhinus canicula. Black lines represent the median estimates of female effective population size, while grey lines the upper and the lower 95% highest posterior density limit.

Raja miraletus showed a considerably different phylogeographic pattern compared to those of both catsharks and the congeneric species, R. clavata. In contrast, the thornback ray (R. clavata) did not display well-identified geographical haplotype clusters. Both species showed any specific mtDNA lineage in the Adriatic Sea and this appears different with respect to the R. polystigma, that exhibited a weakly divergent but fixed COI haplotype (Frodella et al., 2016). The present findings for the thornback ray agree with those of Chevolot et al. (2006), in which a relic, unique cytb haplotype was identified for samples from Corsica, the Adriatic and Black Seas. Our findings also coincide with the CR results of Pasolini et al. (2011), in which no geographical clades were identified within the Mediterranean. Due to a higher adult dispersal potential as suggested by a larger body size, R. clavata may be able to inhabit deeper waters and may thus be able to colonize different geographical areas compared to coastal, shallow-water species, such as the small-sized congeneric R. miraletus (Sion et al., 2004; Serena, 2005; Serena, Mancusi & Barone, 2010).

Although mtDNA variance was not significantly different between Western and Eastern Mediterranean populations in any of the elasmobranch species considered, we detected significant differences in the levels of genetic diversity in three out of four taxa. In R. clavata, R. miraletus and G. melastomus, the mtDNA diversity of the Western Mediterranean samples was significantly lower than the Eastern ones, while the Balearic, Ligurian and Tyrrhenian samples revealed the lowest haplotype and nucleotide diversity, especially in R. miraletus. In general, low values of haplotype and nucleotide diversity may indicate evolutionary and ecological processes (i.e., bottleneck or recent founder events). Thus, further analyses based on high-resolution markers, should be conducted in order to explore these scenarios.

Although the Sicilian Channel has not been a significant barrier in the historical process of differentiation in these demersal elasmobranchs, it may currently be acting as an effective barrier limiting dispersal and gene flow between populations inhabiting the Mediterranean sub-basins (Gubili et al., 2014; Kousteni et al., 2015). Similar patterns of genetic differentiation were also recently detected in R. miraletus (Ferrari, 2017) and in R. asterias (Cariani et al., 2017). The role of the Siculo-Tunisian area as a physical/physiological barrier to population connectivity has also been demonstrated for several teleost fish (Kotoulas, Bonhomme & Borsa, 1995; Mattiangeli et al., 2003; Suzuki et al., 2004; Garoia et al., 2007; Debes, Zachos & Hanel, 2008). Differential physiological effects of seawater temperature and salinity discontinuities in the area and the differences between the Western and Eastern Mediterranean (Hopkins, 1985; Coll et al., 2010) are likely to affect the level of early-life stage dispersal of bony fish. However, the magnitude of this oceanographic break is more evident in benthic teleosts (Kotoulas, Bonhomme & Borsa, 1995; Suzuki et al., 2004; Garoia et al., 2007) and in species inhabiting the northern part of Mediterranean (Debes, Zachos & Hanel, 2008).

Complementary to microevolutionary processes, environmental and ecological factors could also have driven the diversification of Mediterranean elasmobranchs and may account for their discordant phylogeographic patterns. The fundamental role of ecological features in the demographic histories of demersal elasmobranchs is further emphasised by the mismatch distribution and the BSP analyses conducted here. Despite the application of a mutation rate (used as prior in the BSP analysis) that has been estimated solely from rajid lineages and with different mtDNA gene fragments (cytb and 16S rDNA; (Valsecchi et al., 2005; Chevolot et al., 2006), past demographic expansions were detected in all investigated species. In particular, synchronous sudden expansions were experienced by the thornback ray, brown skate, and the blackmouth catshark approximately 40,000-60,000 years ago. In contrast, the small-spotted catshark exhibited a constant demographic growth in the last 350,000 years. Recently, Kousteni et al. (2015) utilised mitochondrial COI and estimated a weak decline in population size in the Aegean Sea that has been related to the restricted habitat availability during the Pleistocene. In contrast, the glacial period potentially caused a slight increase of the population size in the Ionian Sea (Kousteni et al., 2015). Middle and Late Pleistocene cycles of glacial and interglacial periods, with related paleoclimatic and sea-level changes, seem to have influenced demographic histories of the northeastern Atlantic and Mediterranean marine fauna as indicated by species- or population-specific demographic expansions between 1.1 and 0.05 MYA (e.g., algae, Hoarau et al., 2007; invertebrates, Luttikhuizen et al., 2003; Stamatis et al., 2004; Caldèron, Giribet & Turon, 2008; vertebrates, Gysels et al., 2004; Aboim et al., 2005; Alvarado Bremer et al., 2005; Charrier et al., 2006; Chevolot et al., 2006; Larmuseau et al., 2009). Among Mediterranean fish populations, more recent expansions (from 350,000 to 50,000 years ago) have affected those of the Atlantic Bluefin tuna, Thunnus thynnus (Alvarado Bremer et al., 2005) and the sand goby, Pomatoschistus minutus (Larmuseau et al., 2009). While remains difficult to identify factors that could have driven the demographic expansion of these benthic marine taxa in the Mediterranean, overlapping demographic expansions may have been caused by similar environmental shifts such as benthic habitat changes at the beginning of the last glacial period (Würm, from 70,000 to 15,000 years ago; (Graham, Dayton & Erlandson, 2003; Liu et al., 2011). However, within this period our data indicate that the Last Glacial Maximum (from 26,500 to 20,000 years ago) did not strongly affect the demographic histories of marine species and populations inhabiting the northeastern Atlantic and Mediterranean ecosystems by blurring the previous Middle and Late Pleistocene demographic expansions (Chevolot et al., 2006; Hoarau et al., 2007; Larmuseau et al., 2009).

Conclusions

The comparative analysis of phylogeographic and historical demographic patterns for the Mediterranean populations of these elasmobranchs reveals that historical phylogeographic breaks have not had a large impact on their differentiation. The minor role of the Sicilian Channel transition area has potentially prevented the complete sorting of haplotype lineages between the Western and Eastern Mediterranean, though sub-basin-specific lineages have been observed among the species. The demographic histories of the four target species are very similar, indicating a recent origin of these populations with the exception of the brown skate, suggesting that may have experienced a different demographic history likely related to past changes in the benthic habitat conditions. Overall, historical barriers to dispersal appear to play a negligible role in the ecology, distribution and genetic diversity of elasmobranch populations in the Mediterranean compared to biotic and abiotic factors.

Supplemental Information

Table S1 Sample and sequence metadata

Sample and sequence metadata detailing the species considered, the sample name and alignment code, the sub-basin, geographical area and sampling location with available coordinates and year of collection. In addition, haplotype GenBank accession numbers are listed for each individual sample and marker. Public records of North-Eastern Atlantic rajid samples for COI marker were retrieved from GenBank (A) Costa et al. (2012); (B) Knebelsberger et al. (2014); (C) Serra-Pereira et al. (2011).

Click here for additional data file.

Table S2 PCR and sequencing primers

PCR and sequencing primers. Primer pairs and annealing temperatures (Ta) used to amplify and sequencing the three mitochondrial DNA fragments in the four elasmobranch species. The annealing temperatures for the PCR reactions are also reported. GM= G. melastomus; SC, S. canicula; RC, R. clavata; RM, R. miraletus.

Click here for additional data file.

Supplemental Information 1 mtDNA alignment of Raja clavata.

Click here for additional data file.

Supplemental Information 2 mtDNA alignment of Raja miraletus

Click here for additional data file.

Supplemental Information 3 mtDNA alignment of Galeus melastomus.

Click here for additional data file.

Supplemental Information 4 mtDNA alignment of Scyliorhinus canicula

Click here for additional data file.

We thank the two anonymous reviewers and Dr. Panagiotis Kasapidis for the criticisms and suggestions that helped to improve the manuscript. We also thank all contributors to this research. In particular, we are grateful to the MEDITS partners for their sampling effort.

Additional Information and Declarations

Competing Interests

Author Contributions

Animal Ethics

Data Availability

The authors declare there are no competing interests.

Alice Ferrari, Victoria Bertucci Maresca and Alessandro Velonà performed the experiments, analyzed the data, prepared figures and/or tables, authored or reviewed drafts of the paper, approved the final draft.

Fausto Tinti conceived and designed the experiments, performed the experiments, analyzed the data, contributed reagents/materials/analysis tools, authored or reviewed drafts of the paper, approved the final draft.

Rita Cannas, Ioannis Thasitis, Filipe Oliveira Costa, Maria Cristina Follesa, Daniel Golani, Farid Hemida, Cecilia Mancusi, Antonello Mulas, Fabrizio Serena, Letizia Sion and Marco Stagioni contributed reagents/materials/analysis tools, authored or reviewed drafts of the paper, approved the final draft.

Sarah J. Helyar contributed reagents/materials/analysis tools, authored or reviewed drafts of the paper, approved the final draft, the native-speaking co-author Sarah Helyar has carried out comprehensive proofreading throughout the manuscript.

Alessia Cariani conceived and designed the experiments, performed the experiments, analyzed the data, contributed reagents/materials/analysis tools, prepared figures and/or tables, authored or reviewed drafts of the paper, approved the final draft.

The following information was supplied relating to ethical approvals (i.e., approving body and any reference numbers):

The sample shark and skate individuals analysed in the present work were obtained from commercial and scientific fisheries. The activity was conducted with the observation of the Regulation of the European Parliament and the Council for fishing in the General Fisheries Commission for the Mediterranean (GFCM) Agreement area and amending Council Regulation (EC) No. 1967/2006. This Regulation is de facto the unique authorisation needed to conduct this type of activities. Dead fish vertebrates are out of scope of “the University of Bologna - Animal Care and Use Committee” since it authorises/not authorises experimental procedures on living vertebrates.

The following information was supplied regarding data availability:

COI haplotype sequences are available in Genbank under accession numbers: JN944484 –JN944518 and MH472624 –MH472630.

NADH2 haplotype sequences are available in Genbank under accession numbers: JQ729000 –JQ729041 and MH472635.

CR haplotype sequences are available in Genbank under accession numbers: JQ729042 –JQ729123.

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
