# Peer review of "Natural history and molecular evolution of demersal Mediterranean sharks and skates inferred by comparative phylogeographic and demographic analyses"

_PeerJ, doi:10.7717/peerj.5560_

## Round 0.1 · original submission · Major Revisions

I do agree with the reviewers #1 and #2 recommendations and critique on data analyses, discussion and conclusions. In particular I am especially uncomfortable with the phylogenetic analyses being based on a software which admittedly «is still under development and should be considered a "beta" version.». I dare to suggest the use of the freely available and extensively used Netowrk (http://www.fluxus-engineering.com/sharenet.htm).

Reviewer 1 ·

Basic reporting

- language very good throughout the manuscript
- background information includes relevant literature
- figures are all relevant and look appropriate
- the hypothesis can be tested using the applied analyses

Experimental design

- the research question is well defined, whether there is a phylogeographic pattern in four elasmobranch species across the Mediterrenean and if the Sicilian Channel provides a strong biogegraphical barrier
- the study is based on a very dense sampling studying a high number of individuals and 3 mitochondrial markers
- methods are explained in detail, but some of those are, in my opinion, not required to test the hypothesis and should be considered with caution (e.g., BSP; see comments in the attached PDF)

Validity of the findings

- the results are robust and are partly congruent with former published studies
- conclusions could be improved, more discussion is needed why the present results differ from other studies on the same species

Additional comments

See attached PDF

Annotated reviews are not available for download in order to protect the identity of reviewers who chose to remain anonymous.

·

Basic reporting

In this article, the authors perform a comparative phylogeographic analysis on two pairs of closely related elasmobranch demersal species (two skates and two catsharks) in the Mediterranean Sea and adjacent Atlantic, by using sequences of three mitochondrial DNA gene fragments. Although the study is quite interesting and new knowledge is presented, the paper has several flaws that need to be addressed.

The English language of the text, although generally good, should be improved and the manuscript would benefit if proofread by a professional. There are several points where the phrasing seems incorrect: for example in line 46 “paired phylogenetically and ecologically related species” sounds somehow incomprehensible (do you mean “pairs of”?); in line 228 you probably mean “few”=a small number and not “a few”=some; in line 300-301 “the events that overturned the planet” sounds as if an asteroid hit the planet and turned it upside down.

The introduction lacks clear scope, is not coherent and the first part as well as lines 32-37 are quite general and not directly related to the findings. I suggest, after you clearly define the aim of the study (see comments on the next section), to rewrite the introduction in order to better justify your study and to give more details on the previous knowledge directly related to your study (e.g. previous findings on elasmonbranch phylogeography, phylogeographic barriers in the Mediterranean). The authors state in lines 43-44 that “the role of the Sicilian Channel as a partial barrier to elasmobranch dispersal rather than a transition/genetic admixture area is still under debate”, without providing any literature. There are also references, like Tsigenopoulos et al. (2003) (line 26), which deal with freshwater fishes and not marines ones, as it is stated in the introduction.

The structure of the article seems to conform to an acceptable format. Figures are relevant to the content of the article, of sufficient resolution, and appropriately described and labelled,although they contain few grammatical errors in the labels (e.g. in Fig2-5, you should write “haplotypes” (in plural), the “Blue dashes” are actually “Blue dots”, and “relative” seems superfluous).
In Fig.1 you should include on the map the names of the sea regions mentioned in the manuscript. In table 2, the geographic area of the south-western Mediterranean (N. African coast) is not the Alboran Sea. The Alboran Sea is between the Straits of Gibraltar and the Almeria-Oran front. Please correct throughout the text and tables. Moreover, in the Adriatic Sea it is included a sample from the NW Ionian Sea.

The raw data (DNA sequences) have been deposited in GenBank (accession numbers are provided in Table 1) and they are already publicly available. However, they do not contain any metadata (sampling site, collection date), are practically useless for other scientists and cannot be used for checking the results. The authors should provide a supplementary table with the metadata of the sequences (species, sample name, sampling site with coordinates if possible, collection date and accession numbers for each gene) and it would be desirable to put this information in GenBank as well.

Experimental design

The research question (i.e. the aim of the study) is not clearly defined, nor the knowledge gap being investigated is clearly described. As a consequence, the aim of the study, as described in the introduction, is not fully concordant with the analysis followed and the discussion. The authors state that their aim is “to assess phylogeographic structures, infer demographic patterns, and testing for common natural histories and environmental/climatic factors driving their evolution” (L59-71). Regarding the first aim, what they actually do is to focus almost exclusively on whether the Sicilian Channel is a barrier to gene flow for these species (absolute, partial?), ignoring other potential barriers, such as the Almeria-Oran front and the hydrographic isolation of the Adriatic Sea. The AMOVA analysis focuses on the East/West subdivision,while no pairwise comparisons (e.g Φsts) are performed. If this is actually their aim, it should be clearly described and explained in the Introduction, although it would limit the scope of the article. Otherwise, they should examine more thoroughly the genetic structure/differentiation of the species and compare the observed patterns.
Regarding the last aim (“testing for common natural histories and environmental/climatic factors driving their evolution”), this is not sufficiently addressed.

The investigation seems rigorous and performed to a high technical & ethical standard, although the sampling scheme misses some areas of the Mediterranean Sea, like the Aegean Sea and the sample sizes are rather small.

Validity of the findings

The Discussion is not well written, the interpretation of the results is sometimes confusing and not very convincing.
More specifically, the results of the phylogeographic analysis sometimes seem not to be properly interpreted and not to be concordant with previous studies on the same species. Especially for S. canicula, the results contradict previous findings (Gubili et al. 2014, Kousteni et al. 2015), which demonstrated strong genetic structure, but the authors do not try to explain this discrepancy (different sampling scheme, inadequate sample size?). Moreover, all haplotypes of S. canicula but one in the current study are region-private, which is an indication of strong genetic differentiation, but it is also quite peculiar and it should be explained (biased sampling scheme, higher resolution of the marker?). The authors should also take into account that they lack samples from the Aegean Sea, which are genetically quite distinct from that of W. Mediterranean.
For R. clavata, the authors claim that their findings are “somehow unexpected since previous studies revealed a relict and unique cytochrome b Mediterranean haplotype (Chevolot et al., 2006)”, without elaborating further. However, the current findings seem to be concordant with that of Chevolot et al. (2006). The only difference is that given the higher variability of the marker used and the more extensive sampling in the Mediterranean, the authors were able get higher resolution and to reveal more, closely-related haplotypes and a couple of small differentiated clades.
In line 277, the authors claim that “rajid skates exhibited sharply different phylogeographical patterns” compared to catsharks. However, R. clavata did not exhibit any strong phylogeographical pattern unlike R. miraletus and the partitioning of variation according to AMOVA was similar to that of S. canicula.
Overall, the authors should re-examine the interpretation of their results and take into account previous findings.

There are points in the discussion that are rather confusing and not very meaningful. In lines 223-226, you refer to dispersal of catcharks “between the northern (larger) or southern (narrower and steeper) coastal shelves of the Mediterranean” which “contribute to the admixture of both catshark populations”. Do you mean populations of both catshark species? Otherwise, which two populations do your refer to and what do you try to explain here?
In lines 251-261, you try to explain why for some teleosts the Sicilian Channel is an effective barrier and not for the four elasmobranch species studied, by implicating seawater temperature and salinity discontinuities that affect the dispersal of early-life stages in bony fish. However, for the great majority of the teleosts of the Mediterranean, the Sicilian Channel is not an effective barrier as well. The few bony fish species that exhibit this east/west divide are mostly benthic (e.g. flat fish), or have relict distribution in the northern parts of the Mediterranean (sprat).
In lines 262-265 you try to relate the discordant phylogeographic patterns of the studied species with the recent phylogenetic origin of catsharks (in which way? Do you also have any evidence for the phylogenetic origin of the two skates? ).

The final conclusions are too vague.

Reviewer 3 ·

Basic reporting

No comment

Experimental design

No comment

Validity of the findings

No comment

Additional comments

This is a very well written and well organized paper. The language is professional and clear throughout. The introduction gives sufficient background and references are appropriate. The structure and length of the paper is adequate for the subject matter and there are relevant hypotheses relating to environmental conditions over time and the phylogeography and historical demography of four target demersal elasmobranch species.
The study is original, the questions are well defined and answered, the methods are appropriate and well described. The data analysis is robust and well presented and the conclusions are well supported by the results.
Overall this paper was a pleasure to read I have nothing to add that would make it better.

---

## Round 0.2 · Minor Revisions

I think the rebuttal and modifications introduced are satisfactory. But as you can see, there are substantial comments remaining from Reviewer 1 (in an annotated PDF). Please address these comments in a revision, and I anticipate Acceptance afterwards.

Reviewer 1 ·

Basic reporting

See "General comments for the author"

Experimental design

No changes

Validity of the findings

No changes

Additional comments

I appreciate the effort that went into the Discussion to better compare and discuss the present findings with the results found in the literature. However, I also find that the authors included so many examples and got lost in many details, for example, identifying particular Aegean haplotypes in specific species, that are not relevant for the overall discussion. This makes it a little bit difficult to read and it is not all needed to discuss the role of the Sicilian Channel. Also, and unfortunately, I had to invest much more time in understanding and rephrasing the Discussion section as the English was not as Professional as in the previous manuscript (see my comments in the attached PDF).

Annotated reviews are not available for download in order to protect the identity of reviewers who chose to remain anonymous.

---

## Round 0.3 · accepted · Accept

The new version satisfies all comments and suggestions previously made.

# Reviewer 1 ·

Basic reporting

No comment

Experimental design

No comment

Validity of the findings

No comment

Additional comments

Thanks for the revision, looks fine to me now!